# CD28 and 41BB Costimulatory Domains Alone or in Combination Differentially Influence Cell Surface Dynamics and Organization of Chimeric Antigen Receptors and Early Activation of CAR T Cells

**DOI:** 10.3390/cancers15123081

**Published:** 2023-06-07

**Authors:** Marianna Mezősi-Csaplár, Árpád Szöőr, György Vereb

**Affiliations:** 1Department of Biophysics and Cell Biology, Faculty of Medicine, University of Debrecen, 4032 Debrecen, Hungary; 2ELKH-DE Cell Biology and Signaling Research Group, Faculty of Medicine, University of Debrecen, 4032 Debrecen, Hungary; 3Faculty of Pharmacy, University of Debrecen, 4032 Debrecen, Hungary

**Keywords:** chimeric antigen receptor (CAR), immunotherapy, cell therapy, immunological synapse, fluorescence correlation spectroscopy (FCS), receptor mobility, receptor clustering, structure prediction, electric cell-substrate impedance sensing

## Abstract

**Simple Summary:**

Chimeric antigen receptor (CAR)-modified T cells have revolutionized the treatment of chemotherapy-resistant lymphomas, but CAR T cell therapy for solid tumors has been disappointing. This study examined the molecular structure, membrane organization, and mobility of HER2-specific CARs containing different costimulatory domains. Clustering behavior and diffusion kinetics of CARs were influenced by the costimulatory domains, proving to be significant predictors of immune synapse formation and early activation. Our results suggest that CAR T cell therapy for solid tumors needs to consider the molecular structure, membrane organization, and mobility of the chimeric antigen receptors alongside the long-term costimulation-based expansion capacity and anti-tumor activity for optimal therapeutic effect.

**Abstract:**

Chimeric antigen receptor (CAR)-modified T cells brought a paradigm shift in the treatment of chemotherapy-resistant lymphomas. Conversely, clinical experience with CAR T cells targeting solid tumors has been disheartening, indicating the necessity of their molecular-level optimization. While incorporating CD28 or 41BB costimulatory domains into CARs in addition to the CD3z signaling domain improved the long-term efficacy of T cell products, their influence on early tumor engagement has yet to be elucidated. We studied the antigen-independent self-association and membrane diffusion kinetics of first- (.z), second- (CD28.z, 41BB.z), and third- (CD28.41BB.z) generation HER2-specific CARs in the resting T cell membrane using super-resolution AiryScan microscopy and fluorescence correlation spectroscopy, in correlation with RoseTTAFold-based structure prediction and assessment of oligomerization in native Western blot. While .z and CD28.z CARs formed large, high-density submicron clusters of dimers, 41BB-containing CARs formed higher oligomers that assembled into smaller but more numerous membrane clusters. The first-, second-, and third-generation CARs showed progressively increasing lateral diffusion as the distance of their CD3z domain from the membrane plane increased. Confocal microscopy analysis of immunological synapses showed that both small clusters of highly mobile CD28.41BB.z and large clusters of less mobile .z CAR induced more efficient CD3ζ and pLck phosphorylation than CD28.z or 41BB.z CARs of intermediate mobility. However, electric cell-substrate impedance sensing revealed that the CD28.41BB.z CAR performs worst in sequential short-term elimination of adherent tumor cells, while the .z CAR is superior to all others. We conclude that the molecular structure, membrane organization, and mobility of CARs are critical design parameters that can predict the development of an effective immune synapse. Therefore, they need to be taken into account alongside the long-term biological effects of costimulatory domains to achieve an optimal therapeutic effect.

## 1. Introduction

After more than three decades of development, chimeric antigen receptor (CAR) modified T cell therapy has emerged as a promising strategy for cancer immunotherapy [1,2]. CARs consist of an extracellular antigen recognition domain fused to an effector domain, the zeta (ζ) chain of the TCR/CD3 complex that induces signaling upon target engagement [3]. These first-generation CARs were able to specifically recognize and kill tumor targets as well as produce interleukin-2 (IL-2) in vitro [3]; however, they are unsuitable for clinical application due to their suboptimal persistence and anti-tumor activity [4]. Therefore, a costimulatory endodomain derived from either CD28 [5] or 41BB [6] has been incorporated into the CAR backbone, bringing forth the second generation of CARs. In preclinical trials, CD28.z CAR T cells have exhibited rapid proliferation and high levels of interferon gamma (IFNγ) secretion upon target recognition [7,8]. It has also been shown that CD28.z CAR T cells rapidly differentiate into central and effector memory phenotypes [9]. In contrast, 41BB costimulation has shown inferior anti-tumor activity but improved persistence and prolonged T cell division [10]. However, in our recent preclinical trial, HER2-targeting CAR T cell products with CD28 costimulation, enriched in effector memory phenotype and showing balanced CD8+/CD4+ proportions, performed best in vitro and in vivo [11].

Clinically, CD19-targeting second-generation CAR T cells have represented a paradigm shift in treating chemotherapy-resistant leukemias and lymphomas. However, the clinical experience with CAR T cells in solid tumor trials has been disappointing [12]. Further strengthening of the receptors by incorporating two-costimulatory elements (third-generation CARs) could not solve this problem; moreover, in a clinical trial targeting a HER2-positive metastatic colon carcinoma with HER2.CD28.41BB.z CARs, a fulminant cytokine release syndrome developed in a 39-year-old patient that shortly led to the patient’s death [13]. Although CAR T cell therapy could be an encouraging approach in solid tumor therapy, ineffectiveness and safety issues are becoming increasingly apparent, suggesting that their further development requires molecular-level optimization. While new information is continually surfacing on the molecular level functioning of TCR-based synapses [14], we still do not completely understand how CAR molecules and CAR-derived immune synapses are organized and function at the submicron level.

In the present study, we performed a systematic, comprehensive analysis of the molecular organization in the T cell membrane of first-, second-, and third-generation HER2-specific CARs encompassing CD28 and/or 41BB costimulation. We studied the antigen-independent self-association, membrane diffusion kinetics, and early activation of the various CAR molecules using state-of-the-art qualitative microscopic techniques. We found that first-generation HER2.z CARs form fewer, large, high-density, immobile clusters in the cell membrane as opposed to third-generation HER2.CD28.41BB.z CARs that are assembled in a high number of small, fragmented, mobile clusters. These combinations of dynamics and topology were both highly effective in forming active immune synapses, contrasting the second-generation CARs that exhibited intermediate mobility and cluster size and were less readily active in the immune synapse. Concomitantly, we found that the diverse membrane organization and mobility of CARs containing different costimulations are coherent with their modeled molecular structure and have functional consequences in that HER2.z CAR T cells bearing large, high-density preformed clusters are the most efficacious during the first day of the tumor cell killing in vitro.

## 2. Materials and Methods

Unless indicated otherwise, all materials were purchased from Sigma-Aldrich (St. Louis, MO, USA).

### 2.1. Cells and Culture Conditions

HEK 293T packaging cells and the human gastric carcinoma derived N87 cell line were acquired from the American Type Culture Collection (Manassas, VA, USA). These cells were cultured in DMEM (Dulbecco’s Modified Eagle Medium) supplemented with 10% Fetal Bovine Serum (FBS), 2 mM GlutaMAX, and antibiotics. JIMT-1 cells, a human breast cancer cell line established at the Cancer Biology laboratory of the University of Tampere, Finland [15], were cultured in a 1:1 ratio mixture of Ham’s F-12 and DMEM media. This culture medium was supplemented with 20% FBS, 300 U/L insulin, 2 mmol/L GlutaMAX, and antibiotics. For the primary human T cells and CAR T cells, RPMI (Roswell Park Memorial Institute) medium was used, supplemented with 10% FBS, 2 mmol/L GlutaMAX, and antibiotics. All of the cell lines and primary cells were maintained at 37 °C in a humidified atmosphere with 5% CO_2_. Regular monitoring was performed to ensure the absence of mycoplasma contamination.

### 2.2. Primary Antibody Preparation and Conjugation

The monoclonal antibody against ErbB2 (ErbB2-76.5) was produced from hybridoma supernatants (ErbB2-76.5, a kind gift from Y. Yarden, Weizmann Institute of Science, Rehovot, Israel) and purified using protein A affinity chromatography. Conjugation of the primary antibody with Alexa Fluor 488 (Thermo Fisher Scientific, Waltham, MA, USA) was carried out according to the manufacturers’ specifications.

### 2.3. Retroviral Transduction of T Cells

To generate retroviral particles, HEK 293T cells were transiently transfected with pSFG retroviral vectors encoding the HER2-CAR [16], the Peg-Pam-e plasmid that contained the MoMLV gag-pol sequence, and the pMax.RD114 plasmid incorporating the RD114 sequence, using the jetPrime transfection reagent (Polyplus, Illkirch, France). The first-generation HER2-specific chimeric antigen receptor backbones consisted of an IgG heavy-chain N-terminal signal peptide, the FRP5-derived HER2-specific single-chain variable fragment (scFv), the IgG1 short hinge (SH), the transmembrane (TM) region of human CD28 and the cytoplasmic domain of human CD3 zeta (HER2.z). Additionally, the second-generation CARs (HER2.CD28.z and HER2.41BB.z) included a CD28 or a 41BB intracellular costimulatory endodomain, and the third-generation CAR (HER2.CD28.41BB.z) included both of these [16]. After 48 h, supernatants containing the retroviral particles were collected. To produce HER2-CAR T cells, human peripheral blood mononuclear cells (PBMCs) were isolated by Ficoll gradient centrifugation. Using non-tissue culture 24-well plates precoated with 1 µg/mL of anti-CD3e (clone: OKT3; Cat.# 14-0037-82, Thermo Fischer Scientific, Waltham, MA, USA) and anti-CD28 (Cat.# MAB342-100, R&D Systems, Minneapolis, MN, USA) antibodies, T cells were then stimulated at a concentration of 0.5 × 10^6^/mL in 2 mL of complete RPMI. On the second day, the cultures were supplemented with 10 ng/mL human interleukin-7 (IL-7; Cat.# 130-095-367, Miltenyi Biotec, Bergisch Gladbach, Germany) and 5 ng/mL human interleukin-15 (IL-15; Cat.# 130-095-760, Miltenyi Biotec, Bergisch Gladbach, Germany). On day 3, T cells were transduced with retroviral particles on plates precoated with 20 μg/mL RetroNectin (Takara Bio, Kusatsu, Japan). During this process, IL-7 (10 ng/mL) and IL-15 (5 ng/mL) were present. T cell expansion was subsequently supported with IL-7 and IL-15. In parallel, non-transduced (NT) T cells activated with OKT3/CD28 were also expanded using the same protocol in a complete RPMI medium supplemented with 10% FBS, 2 mM GlutaMAX, antibiotics, IL-7, and IL-15. The cells were used for further experiments after a 48 h incubation period.

### 2.4. Flow Cytometric Analysis

To confirm HER2-CAR expression, indirect labeling was carried out with a HER2-Fc fusion protein (Cat.# 1129-ER-050, R&D Systems, Minneapolis, MN, USA) and Alexa Fluor 488-conjugated anti-human IgG (Cat.# A-11013, Invitrogen /Thermo Fisher Scientific/, Carlsbad, CA, USA). T cell purity was assessed by staining with an Alexa Fluor 647-conjugated anti-human CD3 antibody (Cat.# 566687, BD Biosciences, San Jose, CA, USA). All markers used for flow cytometric analysis (antibodies and the HER2-Fc recombinant protein) were applied at a final concentration of 10 µg/mL for 10 min on ice. Analysis was performed using a NovoCyte RYB flow cytometer and NovoExpress software v1.2.1 (ACEA Biosciences, San Diego, CA, USA) on at least 10,000 cells per sample.

### 2.5. Western Blot

Chimeric antigen receptor oligomerization was determined by Western blot. HER2.z, HER2.CD28.z, HER2.41BB.z, and HER2.CD28.41BB.z CAR T cells were washed with PBS, and whole cell lysates were prepared in a lysis buffer containing 50 mM Tris-HCl, 150 mM NaCl, 0.1% Triton X-100, 5 mM EDTA and a protease inhibitor cocktail of 2 mM PMSF, 1 mM Sodium orthovanadate, and 1X cOmplete™ Mini Protease Inhibitor Cocktail tablet (Roche, Basel, Switzerland). Cell lysates were resolved on SDS-polyacrylamide gels in either native or reducing loading buffer containing 0.1 mM DTT and blotted to PVDF membranes. Membranes were blocked with 5% milk in tris-buffered saline containing 0.1% Tween20 for 1 h at room temperature and washed in PBS. The membranes were probed with 1 µg/mL mouse anti-human CD3ζ (Cat.# 556366, BD Biosciences, San Jose, CA, USA) antibody overnight at 4 °C. Membranes were washed and probed with a secondary goat-anti-mouse IgG–horseradish peroxidase (2.5 μg/mL, Cat.# A4416, Sigma-Aldrich/Merck) for 1 h at room temperature. Spot intensity was visualized using WesternBright ECL HRP substrate (Advansta, San Jose, CA, USA) and imaged with a FluorChem Q system (ProteinSimple, San Jose, CA, USA). Spot intensities were quantified using the software ImageJ/Fiji v1.53t [17].

### 2.6. Cluster Analysis

HER2.z, HER2.CD28.z, HER2.41BB.z and HER2.CD28.41BB.z CAR T cells were serum-starved for one hour in serum-free RPMI and washed with 4 °C PBS. CAR T cells were labeled in 10 mM glucose-PBS on ice with 5 µg/mL Alexa Fluor 647-conjugated monomeric HER2 extracellular domain (HER2 ECD; Cat.#10004-HCCH, Sino Biological Europe GmbH, Eschborn, Germany) for 10 min, washed with PBS at 4 °C, and resuspended in 10 mM glucose-PBS. 10^5^ CAR T cells were plated on eight-well chambered slides (ibidi, Gräfelfing, Germany) and incubated at 37 °C during the measurement. CAR clusterization was quantitatively analyzed by AiryScan imaging performed with an LSM 880 confocal laser scanning microscope (Carl Zeiss AG, Oberkochen, Germany) equipped with a water immersion objective (C-Apochromat 40×, NA 1.2) and an AiryScan detection unit. Alexa Fluor 647 was excited by the 633 nm line of a He-Ne laser. An MBS 488/543/633 beam splitter separated the fluorescence emission light from the excitation light. AiryScan images were recorded of the apical membrane surface of the cells. ZEN Blue 2.1 software was used to process the acquired datasets. The software processed each of the 32 Airy detector channels separately by performing filtering, deconvolution, and pixel reassignment. Wiener filter deconvolution with a 2D reconstruction algorithm was applied. Consecutive analysis was carried out in ImageJ/Fiji [17]. The pooled frequency distribution of average image intensities followed a lognormal distribution and was used to exclude the occasional outliers from the analysis. After thresholding, clusters were segmented using the watershed function based on flooding simulations [18]. Average cluster number per 10 µm^2^ apical membrane surface, normalized mean intensity of areas inside and outside clusters, individual cluster size, and integrated intensity values were determined.

### 2.7. Analysis of Immune Synapse Activation

A total of 3 × 10^4^ N87 target cells were plated on eight-well chambered tissue culture-treated slides (ibidi, Gräfelfing, Germany) overnight. Following 15 min of co-culturing with 2 × 10^5^ HER2.z, HER2.CD28.z, HER2.41BB.z or HER2.CD28.41BB.z CAR lymphocytes, the cells were fixed in 1% formaldehyde for 10 min at 37 °C and labeled with 10 µg/mL Alexa Fluor 488-conjugated anti-HER2 antibody (ErbB2-76.5) and either Alexa Fluor 647-conjugated mouse anti-human p-CD3-z (Cat.# sc-9975, Santa Cruz Biotechnologies, Dallas, TX, USA) or PE-conjugated mouse anti-human p-Lck (Cat.# 558552, BD Biosciences, San Jose, CA, USA) in PBS containing 0.05 *v*/*v* % Triton-X for 30 min on ice. The cells were first washed with PBS, then in PBS containing 10 µg/mL DAPI and finally washed in PBS again. Cells were mounted in Mowiol to reduce light-induced fading of the fluorophores. Images were taken with an LSM 880 confocal laser scanning microscope (Carl Zeiss AG, Oberkochen, Germany) and analyzed using the software ImageJ/Fiji [17].

### 2.8. Fluorescence Correlation Spectroscopy

Fluorescence correlation spectroscopy (FCS) measurements were performed on live unstimulated CAR T cells incubated at 37 °C immediately after labeling with Alexa Fluor 647-conjugated monomeric HER2 extracellular domain and recording AiryScan images of their apical membrane surface using the same LSM 880 confocal microscope and optical path, except detection was carried out on the 32-element GaAsP array to provide photon counting. Correlator binning was set to 0.2 µs with a maximum correlation time of 1000 s, and measurements of 10 × 10-s runs were recorded at one selected point in the apical membrane surface of each cell. Runs displaying photobleaching or significant deviations from the average correlation curves because of rare events, such as large fluorescence fluctuations caused by aggregates or membrane motion, were excluded from the analysis. The autocorrelation curves across all donors and replicates were averaged, and non-linear fitting was performed.

Fluorescence autocorrelation curves were calculated using ZEN Black 2.3 software, and their ensemble was fitted to a suitable model accounting for relevant photo-physical events and the diffusion of fluorescent particles. Two components were assumed: free 3D diffusion of the Alexa Fluor 647 conjugated monomer HER2 ECD detached from the cells and free 2D diffusion of membrane-embedded CAR species labeled with the Alexa Fluor 647 conjugated monomer HER2 ECD. The autocorrelation function (Equation (1)) included a term accounting for triplet state formation (Equation (2), *G_triplet_*), a forbidden intersystem crossing of the excited state, characteristic of the fluorophore and its environment) in addition to the term accounting for diffusion (Equation (3), *G_diffusion_*). In the triplet term, *T_t_* denotes the fraction of fluorophores in the triplet state, and *τ_t_* is the triplet correlation time. The triplet term was normalized and weighted to exclude it from the number of diffusing molecules. In the diffusion term, *f*_1_ and *τ_d_*_1_ denote the fractional weight and diffusion correlation time of the 3D component, while *f*_2_ and *τ_d_*_2_ those of the 2D membrane component. *S* is the structural parameter indicating the ratio of axial to lateral focus radii in the ellipsoid-shaped detection volume.
(1)$$G\left(\tau \right)=1+G_{triplet}\times G_{diffusion}$$
(2)Gtripletτ=1+Tt×e−ττt1−Tt
(3)$$G_{diffusion}\left(\tau \right)=\frac{f_{1}}{\left(1+\frac{\tau }{\tau_{d1}}\right)\times {\left(1+\left(\frac{\tau }{\tau_{d1}}\right)\times \frac{1}{S^{2}}\right)}^{0.5}}+\frac{f_{2}}{1+\left(\frac{\tau }{\tau_{d2}}\right)}$$

As non-linear fitting procedures yield more reliable results when the number of fitted parameters is low, we performed calibration experiments to determine key instrumental constants, such as the lateral focus radius *ω_r_* in the ellipsoid-shaped detection volume and the structural parameter *S* (Appendix A). We performed four independent FCS measurements using 10 nM Alexa Fluor 647 dye dissolved in distilled water at 25 °C. A model including a one-component free 3D diffusion and triplet state correction was applied. The diffusion correlation time *τ_d_* of Alexa Fluor 647, and the structural parameter *S*, were determined in the model fit and averaged for the independent measurements. The lateral focus radius *ω_r_* was then calculated using the average *τ_d_* of Alexa Fluor 647 according to Equation (4), where *D* denotes the known diffusion coefficient of 325 µm^2^/s [19] of the Alexa Fluor 647 dye.
(4)$$\tau_{d}=\frac{\omega_{r}^{2}}{4D}$$

Next, we measured the diffusion correlation time *τ_dA647HER2_* of the Alexa Fluor 647 conjugated monomer HER2 ECD in control experiments by employing a free 3D diffusion model with triplet state correction, where the structural parameter *S* was fixed to the value calibrated with free Alexa Fluor 647 (Appendix A).

For cellular measurements, the triplet state fraction and relaxation time (Appendix A), the fractional distribution of the 3D and 2D components (Appendix A), and the 2D diffusion correlation times of CAR species were fitted parameters, while the structural parameter *S* and the diffusion correlation time *τ_dA647HER2_* of free Alexa Fluor 647 conjugated monomer HER2 ECD were fixed to the values determined in the control experiments. Finally, the 2D membrane diffusion correlation times of CAR species were converted to diffusion constants using Equation (4) and the *ω_r_* derived from free Alexa Fluor 647.

### 2.9. Structure Prediction

We applied the RoseTTAFold deep learning-based protein structure prediction algorithm to determine the tertiary structure of various domains in the four different CAR constructs [20] and the native CD3ζ protein. Structures of the HER2-specific single-chain variable fragment FRP5, the IgG1 short hinge with the CD28 transmembrane region, and intracellular costimulatory domains with the CD3z effector domains were predicted individually. Confidence scores were determined corresponding to the predicted Local Distance Difference Test (LDDT), a superposition-free evaluation of local distance differences of all atoms in a model [21], using DeepAccNet [22]. The van der Waals surface was visualized in the iCn3D Structure Viewer [23]. In the case of the intracellular CAR domains, the prediction models with the highest CD3z structure homology were chosen for analysis (Appendix A).

### 2.10. Kinetic Analysis of In Vitro Killing

Electric Cell-substrate Impedance Sensing (ECIS Zθ, Applied BioPhysics, Inc., New York, NY, USA) was used to perform the kinetic analysis of in vitro killing mediated by CAR T cells [24,25]. JIMT-1 target cells were grown in 8W10E PET 8-well arrays with gold electrodes at the bottom. The complex impedance spectrum of cells adhered to the electrodes was assessed from 1 Hz to 100,000 Hz. The effector/target cell ratio was set at 1:1. Treatment started after 25 h of incubating the cells on the plate when the impedance of the target cells reached a plateau representing a completely covered cell culture surface which is essential for appropriate comparison of various treatments. CAR T cells were compared co-temporally in technical replicates, and two independent experiments were run. Impedance was monitored for 25h. Averaged traces were normalized to impedance measured at the start of treatment, and then normalized impedances at every time point were normalized to the corresponding value in the NT T cell control.

### 2.11. Statistical Analysis

Statistical analysis was performed using GraphPad Prism 5 (GraphPad Software, Inc., La Jolla, CA, USA). Data are presented as mean ± SD or ± SEM. Two-tailed *t*-test was used for comparison between two groups, and one-way ANOVA with Tukey’s post hoc test was applied to compare three or more groups. For small samples, normality was confirmed using the Shapiro–Wilk test. *p*-values < 0.05 were considered statistically significant.

## 3. Results

### 3.1. Generation of HER2-Specific CAR T Cells

We generated primary human T cells that express various generation HER2-specific CAR constructs by activating PBMCs and transducing them with the CAR-encoding retroviral vectors. Our constructs consisted of an FRP5 scFv-based HER2-specific recognition domain, an IgG1 short hinge, a CD28 transmembrane domain, a CD3z effector endodomain (HER2.z, first-generation) and either CD28 (HER2.CD28.z, second-generation), 41BB (HER2.41BB.z, second-generation) or both (HER2.CD28.41BB.z, third-generation) costimulatory endodomains. Mean transduction efficiency was determined by flow cytometry on day 4 post-transduction. All constructs were stably expressed on T cells with similar transduction efficiencies (Figure 1a,b). The mean expression was 89%, 76%, 85% and 95% for HER2.z, HER2.CD28.z, HER2.41BB.z and HER2.CD28.41BB.z CAR T cells, respectively (Figure 1b).

### 3.2. Chimeric Antigen Receptors from Different Generations Bear Distinct Organizational Properties on the Molecular and Plasma Membrane Level in Unstimulated CAR T Cells

Our study aimed to identify possible correlations between receptor organization and function of first-, second-, and third-generation anti-HER2 CARs. First, we analyzed the oligomerization susceptibility of various CAR constructs in non-stimulated CAR T cells by non-reducing vs. reducing Western blot. We found that approximately 38% of 41BB.z CARs and 60% of CD28.41BB.z CARs form oligomers, compared to less than 10% of .z and CD28.z CARs, while the rest of the receptors are present in dimeric form (Figure 2a,b). Since this method cannot predict the receptor association in CAR T cell membranes, we also examined the receptor distribution by super-resolution AiryScan microscopy. For this purpose, CARs were stained with Alexa Fluor 647 conjugated monomeric HER2 ECD molecules that minimized the probability of receptor cross-linking and consequential internalization. Approximately 600-nm-thin optical slices were imaged from apical membranes, followed by cluster segmentation and analysis (Figure 2c). In this process, we verified that the mean fluorescence intensities of the apical membrane slices, representing the number of receptors, did not differ significantly between the various CAR constructs (Figure 2d). We found that the approximate number of receptors localized in clusters was consistent across the four constructs (Appendix A). However, the HER2.z and HER2.CD28.z CAR dimers were assembled into fewer but larger clusters with high receptor density, in contrast to the 41BB-containing oligomerized receptors that aggregated in more numerous and significantly smaller clusters with lower integrated intensity (Figure 2e–g).

### 3.3. Diffusional Mobility and Oligomerization Potential of Chimeric Antigen Receptors Built with Different Costimulatory Domains Are Correlated with Their Divergent Molecular Structures

Next, we investigated the lateral diffusion kinetics of CARs by fluorescence correlation spectroscopy. Individual fluctuation traces revealed that HER2.z and HER2.CD28.z constructs frequently formed larger clusters that showed up as higher and longer spikes in the record (Figure 3a), as opposed to 41BB-containing constructs, which had overall shorter spikes with smaller amplitudes. While this observation correlates well with the submicron-sized clusters observed in AiryScan images, it is important to emphasize that the two methods reveal two different hierarchical levels of aggregation. Autocorrelation functions calculated from the fluctuation traces could be best fitted by a two-component diffusion model corrected for triplet state (Figure 3b). To fit this model, the 3D diffusion correlation time (τ_A647HER2_) of the free Alexa Fluor 647-conjugated HER2 ECD used for labeling the CAR, and the structural parameter S of the detection volume were determined in independent calibration measurements (Appendix A). From the fit, we obtained the triplet lifetime, the fractions of components, and the 2D membrane diffusion correlation time of the CAR molecules.

Triplet lifetime showed relatively small, although mathematically significant, variations among the various constructs (Appendix A), which are best explained by the different aggregation states and concomitantly changing molecular environment in the CAR clusters. Analyzing the fractional distribution of diffusing components has revealed that constructs bearing CD28 costimulatory endodomains showed a higher free fraction of dissociated ligands (Appendix A), suggesting that the presence of this domain may reduce the binding affinity of the CAR in this particular construct.

Based on the calibrated size of the confocal volume, the 2D membrane diffusion correlation times were translated to diffusion constants of the respective CAR molecules (Figure 3c). These show that the HER2.z construct exhibited significantly slower mobility, whereas the HER2.CD28.41BB.z CAR exhibited significantly faster mobility compared to HER2.CD28.z and HER2.41BB.z CARs (D_HER2_._z_ = 0.31 ± 0.01, D_CD28_._z_ = 0.44 ± 0.01, D_41BB_._z_ = 0.44 ± 0.01, D_CD28_._41BB_._z_ = 0.53 ± 0.02 µm^2^s^−1^). We correlated these findings with the molecular structure of the constructs modeled by the RoseTTAFold deep-learning-based protein structure prediction algorithm (Figure 3d). The predicted models indicated that with the addition of one or two costimulatory endodomains, the CD3z domain is increasingly shifted away from the inner membrane surface and thus away from the position that the ζ-chain normally assumes in the TCR/CD3 complex. This effect could diminish the interaction of the CD3z domain with phosphoinositides in the plasma membrane [26], thereby causing a gradual increase in CAR mobility with the insertion of each costimulatory endodomain. 

The molecular models generated for the CAR proteins also revealed that two out of three cysteines in the 41BB costimulatory domain are expected to form an intramolecular disulfide bridge (Figure 3d); consequently, the third one, being unpaired, is prone to mediate the oligomerization of the 41BB-containing constructs.

### 3.4. Preformed Clusters and High Mobility of CARs Both Benefit Early Activation in the Immune Synapse

The short-term recruitment and activation properties of the constructs were investigated by confocal fluorescence microscopy of phospho-CD3ζ and phospho-Lck signaling molecules in situ (Figure 4a). Quantitative analysis of mean and integrated synaptic intensity revealed that HER2.z and HER2.CD28.41BB.z constructs were significantly more efficient in inducing short-term CD3ζ phosphorylation than HER2.CD28.z or HER2.41BB.z CARs (Figure 4b, top row). The fluorescence intensity indicating integrated Lck phosphorylation followed the same trend, although the differences were not statistically significant (Figure 4b, lower right panel). We also observed that in HER2.z CAR T cells, the pLck signal covered a significantly larger area of the synapse, and therefore the average synaptic intensity of the construct was weaker than that of the third-generation HER2.CD28.41BB.z CAR (Figure 4b, bottom row). Overall, the HER2.CD28.41BB.z CAR exhibited the most compact, highly activated synapses, which can be attributed partly to the smaller but more numerous clusters it has already formed before stimulation, and partly to it having the highest mobility of all the receptors at the sub-cluster/supramolecular level.

### 3.5. First-Generation CARs Induce the Most Robust Target Cell Killing

Finally, different generations of HER2-CAR T cells were co-cultured with HER2-positive JIMT-1 target cells at a 1:1 effector/target cell ratio for 25 h and tumor-specific killing efficiency was determined by Electric Cell-substrate Impedance Sensing. Data show that HER2.z and HER2.CD28.z CAR T cells are immediately involved in tumor elimination, whereas the engagement of HER2.41BB.z and HER2.CD28.41BB.z CAR T cells in tumor killing is delayed during the first two hours of contact with the tumor (Figure 5). Interestingly, even though short-term activation upon first target cell engagement is most pronounced in T cells redirected by non-costimulated .z and third-generation CD28.41BB.z CARs, longer-term cytotoxic efficacy was found to be grossly different for these two species, as the first-generation .z CARs induced the most robust target cell killing (Figure 5, red trace), while CARs containing 41BB costimulatory endodomains (Figure 5, blue and purple traces) performed the worst. Combining these results with the findings on clustering trends and diffusion dynamics, we can conclude that large, high-density preformed clusters of HER2.z and HER2.CD28.z CARs were the most beneficial for tumor elimination during the first 25 h.

## 4. Discussion

CD19-specific second-generation CAR T cells have represented a paradigm shift in treating B cell derived leukemias and lymphomas; however, the clinical experience with CAR T cells targeting solid tumors is controversial [12]. CD28 costimulatory HER2-specific CARs demonstrated efficient tumor eradication in preclinical animal models [11,16,27], but failed to elicit a therapeutic response in clinical trials [2,4,28,29]. Recently, we have shown the limited anti-tumor efficacy of 41BB.z HER2-CAR T cells in a HER2-positive xenograft model [11]. Third-generation CARs have been developed to overcome the limitations of CD28 and 41BB CARs by taking advantage of both costimulatory domains simultaneously. However, instead of bringing forth enhanced cytotoxicity and improved persistence, third-generation CAR T cells developed a lethal adverse event in a clinical trial that targeted a HER2-positive metastatic colon carcinoma [13]. This regrettable case highlights the importance of thoroughly understanding the function of CARs at the molecular level and the organization of CAR-derived immune synapses at the submicron level. To address this issue, we systematically compared the structure, oligomerization state, clusterization, membrane diffusion, early activation and cytolytic effectiveness of the first-, second- and third-generation HER2-specific CAR constructs containing a CD3z effector domain and combinations or lack of CD28 and 41BB costimulus.

First, we showed that the first-generation HER2.z and second-generation HER2.CD28.z CARs mainly form dimers and are organized into fewer but larger clusters with high receptor density. In contrast, a significant fraction of HER2.41BB.z and HER2.CD28.41BB.z CARs are present in the oligomeric state aggregated into smaller membrane clusters. This observation can be explained by the structure prediction models of the constructs suggesting that incorporating a 41BB costimulatory endodomain results in a unique tertiary CAR structure. This is based on the presence of three cysteine residues, two of which are predicted to form an intramolecular disulfide bridge, while the third unpaired cysteine possibly promotes the formation of higher CAR oligomers in an antigen-independent manner. However, this potential mechanism for oligomerization does not exclude the possibility of crosslinking other CAR domains, including the scFv [30]. 

Next, we analyzed the mobility of the various generation CAR molecules by fluorescence correlation spectroscopy. Fitting the ensemble autocorrelation function derived from individual fluctuation traces with the appropriate diffusion model showed that the HER2.z construct, which often assembles into larger clusters, was significantly slower, whereas the HER2.CD28.41BB.z CAR showed significantly faster diffusion in the T cell membrane than the second-generation HER2.CD28.z or HER2.41BB.z constructs. Since the RoseTTAFold deep-learning-based algorithm predicted that the insertion of one or two costimulatory domains gradually removes the CD3z effector domain from the inner surface of the cell membrane, we propose that this could explain why the lateral diffusion of these chimera receptors also accelerates proportionally with the insertion of costimulatory domains. In T cells, the cytoplasmic tail of the native CD3ζ was shown to selectively form a complex with several phosphoinositides in the plasma membrane [26], and the inhibition of this interaction was reported to significantly increase the lateral diffusion rate of the TCR complex prior to ligand engagement [31]. Our results suggest that similarly to the TCR, the diffusion dynamics of CAR constructs could also be altered by the ability of the CD3z effector domain to interact with phosphoinositides in the inner leaflet of the cell membrane, thereby causing a gradual increase in CAR mobility with the insertion of every costimulatory domain displacing the native ζ chain from its usual position.

Balagopalan and colleagues have recently shown that upon stimulus, the formation of native TCR nanoclusters in the resting T cell membrane improves signal transduction [32]. Based on this result, we hypothesized that the distinct qualities of preformed receptor assemblies in the unstimulated CAR T cell’s membrane, as well as the diffusional mobility of the chimeric antigen receptors, could affect the formation of the immunological synapse and early activation signaling events upon engagement with the specific tumor target. The short-term recruitment and early activation properties of the constructs were investigated by confocal microscopy, exploiting phospho-CD3ζ and phospho-Lck specific labeling. Quantitative image analysis showed that HER2.z and HER2.CD28.41BB.z constructs induced stronger short-term CD3ζ and Lck phosphorylation than HER2.CD28.z and HER2.41BB.z CARs, indicating that both the preformed high-density clusters of HER2.z dimers and the fast membrane diffusion of HER2.CD28.41BB.z oligomers can be beneficial for early activation. However, regarding the prolonged cytotoxic effect of CARs in the first 25 h of co-culture with target cells, the HER2.CD28.41BB.z and HER2.41BB.z CARs were the least effective, while the HER2.z CARs exhibited the most efficient cytolytic activity. This cautions that even though both pre-clustered and highly mobile CARs can be very effective in early immune synapse formation, stably aggregated larger CAR clusters may allow better performance in sequential killing. Additionally, while the efficacy of immune synapse formation and the kinetics of early signaling events in the time frame of minutes may largely depend on the characteristics of preformed CAR membrane structures and their mobility, during the first few hours of activation the distinct effects of the costimulatory domain’s specific signaling pathways also become apparent. Particularly, 41BB was found to mediate a noncanonical NF-κB signaling unique to this type of costimulation, promoting long-term CAR T cell survival and persistence, but correlating with reduced proximal signaling as shown by Zap-70 phosphorylation, which was found to be approximately 1.2-fold greater in .z CAR T cells compared to 41BB.z CAR T lymphocytes after 12 h of activation [33].

## 5. Conclusions

Our results from molecular and functional imaging suggest that chimera antigen receptors incorporating 41BB are homooligomerized and that these oligomers form a larger number of smaller sized clusters in the membrane of resting T cells. Lateral diffusion of CARs decreases proportionally as the CD3z effector domain is positioned further away from the inner surface of the membrane, owing to the insertion of one or more costimulatory domains. This suggests that the interaction of CD3z with phosphoinositides seen in the native TCR/CD3 complex also takes place in chimeric receptors with the CD3z effector domain. This interaction is of potential importance, since the slowest .z CARs, which form large receptor clusters, induce a rapid and potent CD3ζ phosphorylation upon the formation of immune synapses. However, the high mobility CD28.41BB.z CARs that feature preformed oligomers also induce a similarly efficient CD3ζ phosphorylation, both being superior to CD28.z and 41BB.z CARs in this respect. Furthermore, the synapses formed by CD28.41BB.z are more compact, with more pronounced Lck recruitment and phosphorylation. Overall, this suggests that greater mobility and preassembled receptor oligomers or larger clusters may all promote the formation of active immune synapses.

Interestingly, although short-term activation at the first target cell adhesion is most pronounced for T cells redirected by non-costimulated .z and third-generation CD28.41BB.z CARs, the longer-term cytotoxic efficacy for these two species is very different; impedance-based real-time monitoring of target cell viability over 25 h indicates that first-generation .z CARs provide the most enduring and efficient sequential target cell killing for T cells, while CARs with 41BB costimulation perform the worst. This, however, may well be attributed to downstream signaling from the immune synapse and consequential regulatory changes.

Cumulatively, our observations underline the importance of receptor structure and mobility, the tendency to spontaneously form oligomers and the ability to engage in membrane-proximal molecular interactions as design parameters predicting efficient synapse formation upon first target engagement. However, there is a caveat that even if these principles are observed, the quality of costimulation becomes a critical determinant of CAR T cell efficacy in the first several hours of effector–target co-culture.

## Figures and Tables

**Figure 1 cancers-15-03081-f001:**
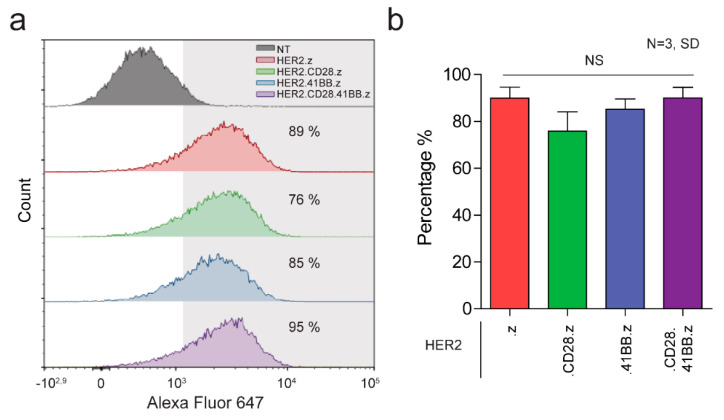
Generation of CAR T cell products from first to third generation. (**a**) Representative flow cytometry histogram of non-transduced (NT) T cells, and HER2.z, HER2.CD28.z, HER2.41BB.z and HER2.CD28.41BB.z CAR T cells. (**b**) Expression of the HER2-CARwas quantified with indirect labeling using a HER2-Fc fusion protein and an Alexa Fluor 647-conjugated anti-human IgG (N = 3). Histograms represent mean ± SD.

**Figure 2 cancers-15-03081-f002:**
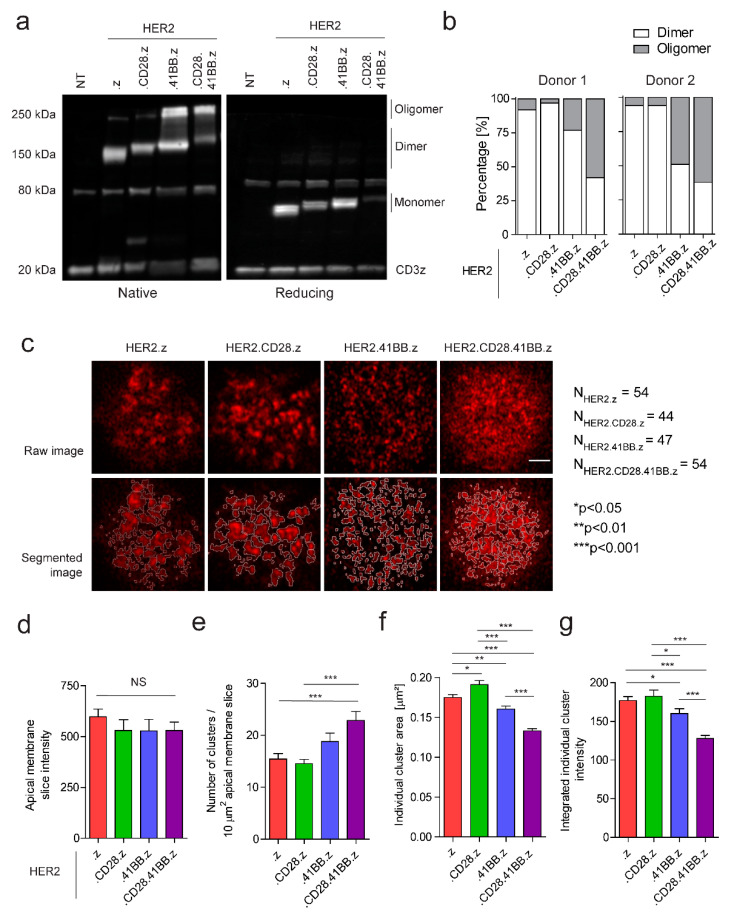
Assessing the oligomerization state and surface organization of various chimeric antigen receptors in the unstimulated CAR T cell membrane. (**a**) Western blot analysis of HER2.z, HER2.CD28.z, HER2.41BB.z and HER2.CD28.41BB.z CAR T cells. The membranes were probed for anti-human CD3ζ. (**b**) Spot intensities were quantified using ImageJ/Fiji, and the proportion of dimer/oligomer states was determined. (**c**) Representative raw and segmented AiryScan images of fluorescently tagged CARs. Clusters were segregated by watershedding and quantified. The scale bar represents 2 µm. (**d**) Total intensity of the apical membrane slices. (**e**) Average number of clusters per 10 µm^2^ apical membrane surface. (**f**) Distribution of individual cluster areas. (**g**) Integrated intensity of individual clusters. The charts represent mean ± SEM; nHER2.z = 54, 4 donors; nHER2.CD28.z = 44, 3 donors; nHER2.41BB.z = 47, 4 donors; nHER2.CD28.41BB.z = 54, 4 donors; * *p* < 0.05, ** *p* < 0.01, *** *p* < 0.001.

**Figure 3 cancers-15-03081-f003:**
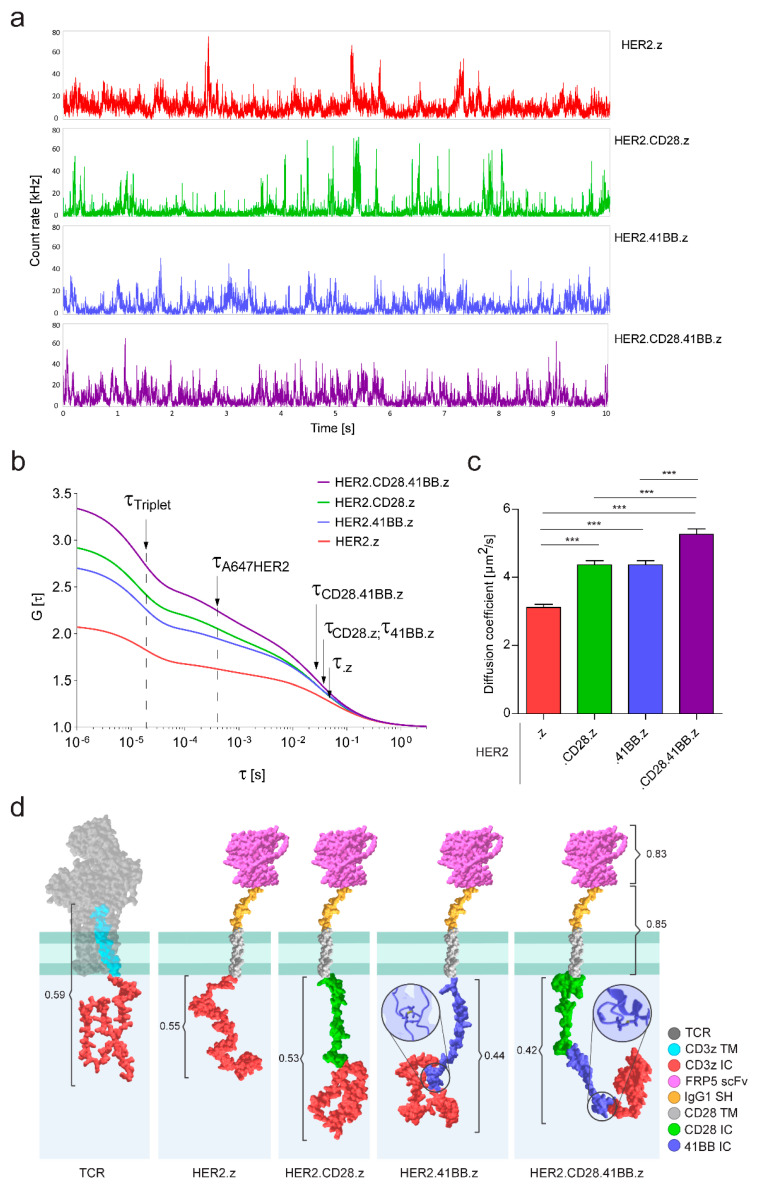
Membrane diffusion analysis and tertiary structure modeling of HER2-specific chimeric antigen receptors. (**a**) Representative fluorescence fluctuation intensity traces of 10-s FCS runs. (**b**) Pooled autocorrelation curve for each CAR. Position of triplet lifetime (τ_Triplet_), diffusion correlation times of the detached Alexa Fluor 647 conjugated monomer HER2 ECD (τ_A647HER2_) and the various CAR species (τ._z_; τ._CD28_._z_; τ._41BB_._z_; τ._CD28_._41BB_._z_) are indicated by arrows. (**c**) Diffusion coefficients of CAR constructs. The chart represents mean ± SEM, nHER2.z = 300, 4 donors; nHER2.CD28.z = 260, 3 donors; nHER2.41BB.z = 276, 4 donors; nHER2.CD28.41BB.z = 289, 4 donors; *** *p* < 0.001. (**d**) Schematic diagram illustrating the tertiary structure of the native TCR and the CAR constructs. TCR: T Cell Receptor complex extracellular domains; CD3z TM: CD3ζ transmembrane segment; CD3z IC: CD3ζ intracellular domain; FRP5 scFv: single chain variable fragment of FRP5 antibody targeting the HER2 ECD; IgG1 SH: short hinge linker derived from IgG1; CD28 TM: transmembrane segment derived from CD28; CD28 IC: costimulatory domain from CD28; 41BB IC: costimulatory domain from 41BB. Confidence scores of the predicted models and intramolecular disulfide bonds are indicated.

**Figure 4 cancers-15-03081-f004:**
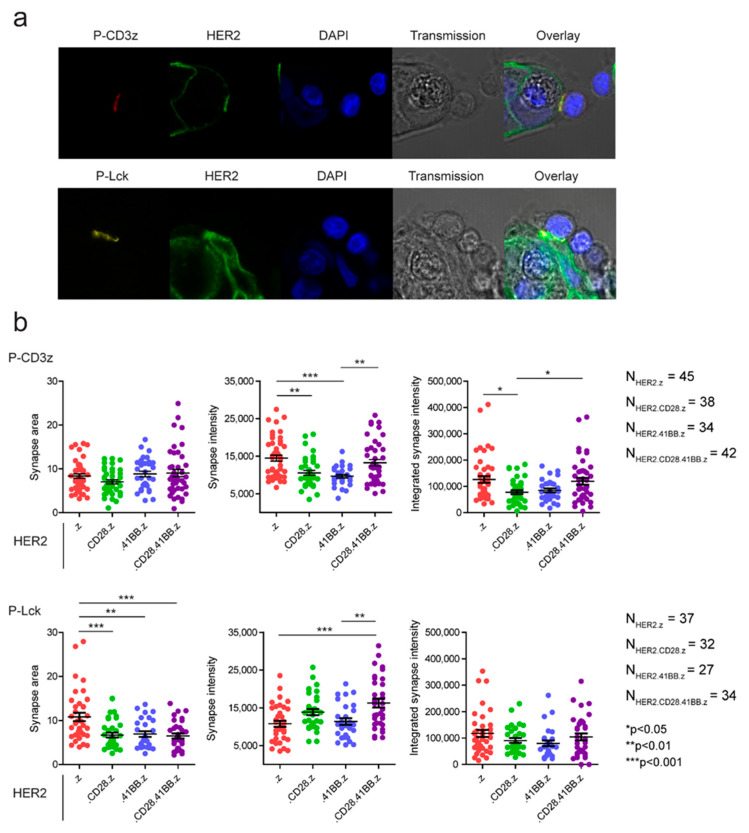
Early activation signaling events were determined by measuring CD3ζ and Lck phosphorylation. (**a**) Representative microscopic images following 15 min of effector–tumor co-culturing. (**b**) Quantitative image analysis of CD3ζ and Lck phosphorylation. The area of the immune synapse, the average and the integrated intensity of the phospho-specific label in the synapse are plotted showing the distribution of individual data points as well as their mean ± SEM. * *p* < 0.05, ** *p* < 0.01, *** *p* < 0.001.

**Figure 5 cancers-15-03081-f005:**
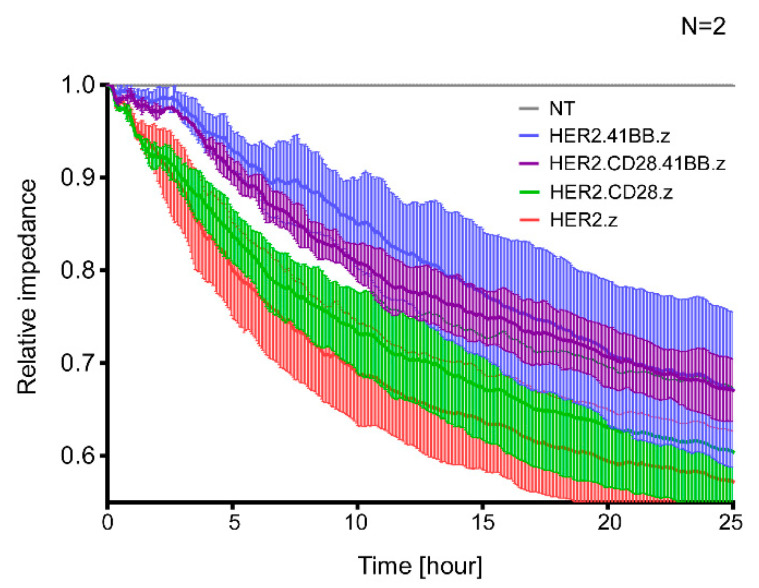
Kinetic analysis of in vitro killing of the various CAR T cells. Killing of JIMT-1 target cells was determined using the Electric Cell-substrate Impedance Sensing method. The effector/target ratio was 1:1. Impedance was monitored for 25 h. Averaged traces were normalized to impedance measured at the start of treatment, and these normalized impedances were normalized to the corresponding value of the non-transduced (NT) T cell control at every time point. The graph represents mean ± SD from N = 2 independent experiments with 2 technical replicates.

## Data Availability

The data presented in this study are available in this article and its Appendix A.

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
