# Peer review of "CD28 and 41BB Costimulatory Domains Alone or in Combination Differentially Influence Cell Surface Dynamics and Organization of Chimeric Antigen Receptors and Early Activation of CAR T Cells"

_cancers, 2023, doi:10.3390/cancers15123081_

Round 1

Reviewer 1 Report

Incorporation of CD28 or 41BB costimulatory domains into CARs in addition to the 26 CD3z signaling domain is known to improve the long-term efficacy of T cell products. However, their influence on early tumor engagement has not been reported.W The authors report studies of the antigen-independent self-association and membrane diffusion kinetics of first, second, and third generation HER2-specific CARs in the resting T cell membrane using super-resolution AiryScan microscopy and fluorescence correlation spectroscopy, in correlation with RoseTTAFold-based structure prediction and assessment of oligomerization in native Western blots.

Their data argues in support of the hypothesis that the molecular structure, membrane organization, and mobility of CARs should be considered critical design parameters that can enable prediction of the development of an effective immune synapse. Accordingly, they argue that these variables should be taken into account, along with the long-term biological effects of costimulatory domains, in order to optimize the  therapeutic effect of the CAR-T cells. Overall an intriguing artcle to read, and should certainly be published!

No major concerns

Author Response

It was a pleasure to read that you found this work intriquing and publishable. Thank you very much for your kind words.

Reviewer 2 Report

The authors analyzed design parameters of HER2-specific CARs to predict the efficiency of synapse formation. They have used the latest and highly specialized microscopy methods to determine oligomerization and clusterization. They also analyzed synapse activation, performed structure prediction, and kinetics of killing of target cells.

In solid tumors, there are still some obstacles to overcome before CAR can be successfully used. Understanding the molecular organization in the T cell membrane of CARs will contribute to being able to safely use CARs in solid tumors.

The authors included first, second and third generation CARs in their study. The results are adequately and comprehensively presented. Figure 1 shows that transduction efficiency of all transduced T-cells was comparable. The constructs differ in oligomerization susceptibility and in clustering. Results shown in Figure 2 are appropriately explained in the legend and in the text.

I do not understand the meaning or origin of triplet state (from line 225 in the methods in 3.3 in the results). Probably, the authors could explain the concept in more detail?

Figure 3d is an informative plot explaining the findings of this study and the effects of the structural differences (e.g. the 3 cysteines or the increasingly distance of the CD3z domain).

Differences in activation of the synapsis are shown in 3.4.

Observations regarding target cell killing agree with literature. Findings of this study help to explain these observations. Do the authors have further explanations why short-term activation is similar for .z and third generation, but longer-term cytotoxicity is not (line 434)?

What does N=2 in Figure 5 refers to? Two independent experiments? Does the confidence interval in the figure originate from the technical replicates?

The results are well explained and classified in the discussion.

In summary, this work is an important contribution to make CARs applicable to solid tumors.

Author Response

Thanks you very much for the positive assessment of our work, and acknowledging its potential importance in the CAR T therapy of solid tumors. Also, we appreiate the questions which have allowed us to improve the manuscript. Please find the the answers herebelow.

I do not understand the meaning or origin of triplet state (from line 225 in the methods in 3.3 in the results). Probably, the authors could explain the concept in more detail?

Thank you for asking this. The photophysical process of triplet formation arises from a forbidden transition called intersystem crossing from the excited singlet state to the so-called triplet state. For conventional fluorescent molecules this state persists for a duration of 1-5μs. Since in this state there is no fluorescence emission, there is, so to say, an absence of photons that signal the diffusion of the molecule through the observation volume proportional to the fraction of total fluorophores in the triplet state in any moment and the time spent there. This shows up as an increase in the correlation amplitude, thereby deviating upwards from the flattened curve at shorter correlation times. This deviation can be modeled and fitted so that the diffusing fluorescing components can be correctly quantitated. It is characteristic of the fluorophore used and its immediate environment, and not important for the interpretation of diffusion properties beyond taking it mathematically into account. We have added a brief reference related to this to the methods section.

Do the authors have further explanations why short-term activation is similar for .z and third generation, but longer-term cytotoxicity is not (line 434)?

We really appreciate this proposition. In fact, while the efficacy of immune synapse formation and the kinetics of early signaling events in the time frame of minutes may largely depend on the characteristics of preformed CAR membrane structures and their mobility, during the first few hours of activation the distinct effects of the costimulatory domain’s specific signaling pathways also become apparent. Particularly, 41BB was found to mediate a noncanonical NF-κB signaling unique to this type of costimulation, promoting long term CAR T cell survival and persistence, but correlating with reduced proximal signaling as shown by Zap-70 phosphorylation, which was found to be approximately 1.2-fold greater in .z CAR T cells compared to 41BB.z CAR T lymphocytes after 12 hours of activation (PMC7883633). We have added this to the discussion.

What does N=2 in Figure 5 refers to? Two independent experiments? Does the confidence interval in the figure originate from the technical replicates?

Yes, N=2 refers to two independent experiments, both having two technical replicates. We thank you for noticing the lack of this information and have added this to the figure legends.